# DecayNet: A Study on the Cell States of Long-Short Term Memories

## Abstract

It is unclear whether the extensively applied long-short term memory (LSTM) is an optimised architecture for recurrent neural networks. Its complicated design makes the network hard to analyse and non-immediately clear for its utilities in real-world data. This paper studies LSTMs as systems of difference equations, and takes a theoretical mathematical approach to study consecutive transitions in network variables. Our study shows that the cell state propagation is predominantly controlled by the forget gate. Hence, we introduce DecayNets, LSTMs with monotonically decreasing forget gates, to calibrate cell state dynamics. With recurrent batch normalisation, DecayNet outperforms the previous state of the art for permuted sequential MNIST. The Decay mechanism is also beneficial for LSTM-based optimisers, and decrease optimisee neural network losses more rapidly.

## 1 Introduction

Neural networks are powerful universal approximators that are difficult to interpret. This paper presents a numerical study on the forward pass of the *long-short term memory* (LSTM) (Hochreiter & Schmidhuber, 1997) recurrent neural networks (RNNs). We treat LSTMs as systems of difference equations and present visualisations to study and to clarify consecutive transitions in the cell state dynamics. We introduce **DecayNets** as an addition to the LSTM architecture – DecayNets have monotonic decays in forget gates and enjoy smooth transitions in their cell state values.

LSTMs are versatile models that have been used to advance the state-of-the-art for a variety of machine learning problems. This covers handwriting recognition (Graves et al., 2009), speech recognition (Sundermeyer et al., 2012), and text modelling (Kim et al., 2016). There are an abundance of work that extends on the basic LSTM architecture. Popular extensions include the attention mechanism (Bahdanau et al., 2014) and the bidirectional design (Schuster & Paliwal, 1997). It is also not uncommon to blend LSTMs into vision tasks to expand network practicalities (Xu et al., 2015).

However, it is unclear whether LSTMs are optimal architectural designs for RNNs (Jozefowicz et al., 2015). Additionally, the significance of their individual components are unclear, and empirical explorations are required for understanding their utility in real-world data (Karpathy et al., 2015). This lack of interpretability limits our ability in designing better and more *transparent* networks.

RNNs repeatedly integrate new observations into implicit hidden variables to model temporal representations of sequences. Like dynamical systems, the network undergoes iterative computations on a same set of operators. The transitional dynamics can be visualised through cobweb diagrams (Glendinning, 1994); and numerical analyses allow us to predict *shrinkage* and *growth* in variables.

We show that the updating scheme of the LSTM cell state volatilely alternates between a *catch* and a *release* phase. The former controls the shrinkage while the latter controls the growth of cell state magnitudes. Our study reveals that the forget gate dictates the dynamical alterations of the cell state. We hence introduce DecayNets, LSTMs with monotonically decreasing forget gates, for stabilising cell state propagation and for making networks more interpretable.

With recurrent batch normalisation (Cooijmans et al., 2016), DecayNets outperform previous state of the art results (best: 97.8%; mean: 97.4%) on the fixed-length permuted pixel-by-pixel (sequential) MNIST task (Le et al., 2015; LeCun et al., 1998). For the arbitrary-length learning-to-learn task (Andrychowicz et al., 2016), LSTM-optimisers with the Decay mechanism decrease multi-layer perceptron (MLP) losses more rapidly, and converge to lower and less varied MLP losses.

## 2 THE VANILLA LSTM

DecayNets will be frequently compared to conventional LSTMs. The most commonly implemented LSTM RNN was proposed by Graves & Schmidhuber (2005), and this paper will refer to the architecture therein that paper as *the* LSTM.

LSTMs are recursive systems driven by its input $\boldsymbol{x}_t$, which has the data-size dimensionality $M$. The network propagates for a data-length of $D$ units of time, with the subscript $t$ as the recursive instance, *i.e.*, $t = 1 \cdots D$. Each instantaneous computation (in abbreviated form) encompasses

**System (1):** the vanilla LSTM

the gated variables: $\quad \boldsymbol{f}_t, \boldsymbol{i}_t, \boldsymbol{o}_t = \quad \sigma(\mathbf{W}_{\{F,I,O\}}\boldsymbol{x}_t + \mathbf{W}_{R\{F,I,O\}}\boldsymbol{q}_{t-1} + \boldsymbol{b}_{\{F,I,O\}})$,

the internal input: $\quad \boldsymbol{a}_t = \quad \tanh(\mathbf{W}_A\boldsymbol{x}_t + \mathbf{W}_{RA}\boldsymbol{q}_{t-1} + \boldsymbol{b}_A)$,

the cell state: $\quad \boldsymbol{s}_t = \quad \boldsymbol{f}_t \odot \boldsymbol{s}_{t-1} + \boldsymbol{i}_t \odot \boldsymbol{a}_t$,

the hidden state: $\quad \boldsymbol{q}_t = \quad \boldsymbol{o}_t \odot \tanh(\boldsymbol{s}_t)$,

with

the dimensionality: $\quad \boldsymbol{x}_t \in \mathbb{R}^M, \quad \boldsymbol{f}_t, \boldsymbol{i}_t, \boldsymbol{o}_t, \boldsymbol{a}_t, \boldsymbol{q}_t, \boldsymbol{s}_t, \boldsymbol{b}_{\{F,I,O,A\}} \in \mathbb{R}^N, \quad$ and

$\quad \mathbf{W}_{\{F,I,O,A\}} \in \mathbb{R}^{N \times M}, \quad \mathbf{W}_{R\{F,I,O,A\}} \in \mathbb{R}^{N \times N}$.

The input $\boldsymbol{x}_t$ and the hidden state $\boldsymbol{q}_t$ contribute to other variables via $\mathbf{W}$s and via $\mathbf{W}_R$s. The dimensionality $N$ is user-defined and $\odot$ is the operator for element-wise product. Along with biases $\boldsymbol{b}$s, the regressive contributions are activated by either the sigmoidal function $\sigma$ or the hyperbolic tangent function $\tanh$. During the update of the cell state $\boldsymbol{s}_t$, the forget gate $\boldsymbol{f}_t$ relinquishes fragments of its previous instance $\boldsymbol{s}_{t-1}$, and the input gate $\boldsymbol{i}_t$ fine-tunes replenishment from the internal input $\boldsymbol{a}_t$. Finally, the variable $\boldsymbol{q}_t$, which also serves as the output of the network, is a transformation of $\boldsymbol{s}_t$ synchronised with the output gate $\boldsymbol{o}_t$.

## 3 LSTMs AS DIFFERENCE EQUATIONS

Difference equations are recurrent relations with discrete propagation; and the term dynamic refers to the dependency between time and geometric space. Dynamical behaviours of a difference equation can be studied to determine whether variables will grow or shrink. This allows us to explain behaviours of individual LSTM variables and clarify network mechanics.

This section discusses three key concepts – the importance for studying the LSTM cell state; the two dynamic regimes for cell state values to grow and to shrink; and that the forget gate can be controlled to yield interpretable and predictable cell state motions. Cobweb diagrams will be used to support concept visualisations, and a brief introduction to the technique can be found in **Appendix A**.

### 3.1 THE CELL STATE REVISITED

This paper postulates the cell state as the source of LSTMs' modelling capability. Furthermore, we regard gated variables as secondary variables. Our justifications are as follows.

First, gated variables only exist to read and to set the hidden variables of $\boldsymbol{q}_t$ and $\boldsymbol{s}_t$. That is, they exist to assist LSTMs to memorise better, but are not a part of the memories. Second, $\tanh$ is a one-to-one function, thus dynamic properties of $\boldsymbol{q}_t$ are inherited from $\boldsymbol{s}_t$. Last, $\boldsymbol{a}_t$ is included in the update of $\boldsymbol{s}_t$, and acts as a small component to the grander dynamic entailed in $\boldsymbol{s}_t$. For these reasons, we postulate the cell state as the most important variable and study it as a difference equation.

### 3.2 A PIECEWISE ANALYSIS ON THE CELL STATE

We study the cell state, $\boldsymbol{s}_t = \boldsymbol{f}_t \odot \boldsymbol{s}_{t-1} + \boldsymbol{i}_t \odot \boldsymbol{a}_t$, separately as $\boldsymbol{s}_p = \boldsymbol{f}_t \odot \boldsymbol{s}_{t-1}$ and $\boldsymbol{s}_q = \boldsymbol{i}_t \odot \boldsymbol{a}_t$. Since we have asserted gated variables as secondary variables, we can think of $\boldsymbol{f}_t$ and $\boldsymbol{i}_t$ as constant vectors with elements between $0$ and $1$. That is, individual dimensions of $\boldsymbol{s}_p$ and $\boldsymbol{s}_q$ have cobweb outlines that are analogous to **Figures 1(a)** and **1(b)** respectively.

For the rest of this paper, we will refer to the marginal dimensions of $\boldsymbol{f}_t, \boldsymbol{i}_t, \boldsymbol{a}_t, \boldsymbol{x}_t$, and $\boldsymbol{s}_t$ as $f_{mt}$, $i_{mt}, a_{mt}, x_{mt}$, and $s_{mt}$, respectively. For instance, each dimension of $\boldsymbol{s}_q$ is $i_{mt}\,a_{mt}$ and inherits the diverse characteristic outline of **1(b)** through $\tanh$ from $a_{mt}$.

Consecutive transitions in network variables are shown as the dotted trajectories in **Figure 1**. They start from the rhombuses and ends on the circles. The characteristic outlines of the sub-dynamics are the solid lines, and the dashed lines represent the $45$-degree $y = x$ lines.

The two subfigures address qualitatively different dynamic regimes. The origin of **Figure 1(a)** is a sink that attracts propagation trajectories and decreases variable magnitudes. Conversely, the origin of **1(b)** is a source that repels propagation trajectories and increases variable magnitudes. We refer to **1(a)** as the dynamic catch regime; and refer to **1(b)** as the dynamic release regime.

The update of the LSTM cell state is thus a composition of two opposing modifiers. With one that dedicates to its growth, and the other specialised in its shrinkage. It is thus crucial to understand which one, between $s_p$ and $s_q$, serves as the more dominant evolution component. We will then reformulate the more dominant component with deterministic properties to redefine the network with interpretable and affirmative qualities.

A key difference between the opposing regimes of **Figure 1** is the boundedness of the characteristic outlines. The sub-dynamic of $s_q$ is limited between $\pm 1$, while that of $s_p$ is unbounded. This implies that $s_q$ contributes less to the additive relationship of $s_k$, and is less impactful than $s_p$. With the dominant component identified, our next target is to understand how $s_p$ affects vanilla LSTMs. The insight will allow us to propose sensible modifications to make LSTM mechanics more interpretable.

### 3.3    THE FORGET GATE AND THE CATCH-AND-RELEASE DYNAMICS

Based on the two insights of, first, $s_p = f_t \odot s_{t-1}$ serving as the dominant modifier to $s_t$, and second, as each of its dimensional outline $s_{m(t-1)}$ being modified by the gradient $f_{mt}$, we conjecture that the forget gate controls the *overall* gradient of the characteristic outline of the cell state. That is, $f_t$ predominantly controls the stability of the origin, and dictates the dynamic alterations between the catch and the release phases. As a consequence and regardless of $s_q$, values of the cell state are more likely to grow with large $f_t$, and more likely to shrink with small $f_t$.

We use **Figure 2** to support these inferences. The first row simulates the near-random dynamics of a vanilla LSTM cell state. We present rows two and three to show stark contrasts, and advantages, of generating predictable cell state propagation with controllable forget gate values.

### 3.3.1    THE NEAR-RANDOM DYNAMICS OF A VANILLA LSTM CELL STATE

Gated units of LSTMs are not hardwired with deterministic propagation. As a result, they evolve in a near-chaotic fashion. The first row of **Figure 2** simulates behaviours of $s_t$ under unregulated $f_t$.

The three subfigures of the row serves as consecutive instances of a propagation, *i.e.*, $t = [1 \ldots 3]$. The presentation follows **Figure 1**, and dot-dash lines are appended to instances $t = 2$ and $3$ to show the characteristic outlines of their previous instances. Parameters $f_{m(1\ldots3)} = [1, 0.25, 1]$ and $i_{m(1\ldots3)} = [1, 0, 0.75]$ are carefully chosen to exaggerate impacts brought to the vanilla LSTM cell state by the near-chaotic evolution of gates. Real scenarios and experimental set-ups that justify the choice of these demonstrative parameters can be found in **Appendix B**.

These settings lead to sudden catches and abrupt releases. The disordered dynamics make it difficult to predict growth and shrinkage in cell state magnitudes. It is also unclear how such drastic behaviours can assist us to understand the learned features of a trained LSTM.

### 3.3.2    INTERPRETABLE CELL STATE DYNAMICS UNDER CONTROLLED FORGET GATES

Let us now consider cell state dynamics under controlled $f_t$s. The second row has $f_{mt} = 0.9$ with $i_{m(1\ldots3)} = [1, 0.75, 0.5]$; and the third row has $f_{mt} = 0.25$ with $i_{m(1\ldots3)} = [1, 0.75, 0.15]$. Consistent growth and consistent shrinkage are observed in rows two and three respectively.

The magnitudes evolve monotonically because the controlled forget gates expose cell states to homogeneous dynamic regimes. Large constant $f_{mt}$s expose the cell state to consistent (*global*) releases; whereas small constant $f_{mt}$s expose $s_{mt}$ to *global* catches.

Dynamic reversals may occur in $s_t$ through $s_q$, but only under the unlikely incidents where combinations of extreme values of $i_t$ and $a_t$ are simultaneously presented. In addition, the dynamic reversals are merely temporary (*local*) due to the boundedness of values in $s_q$.

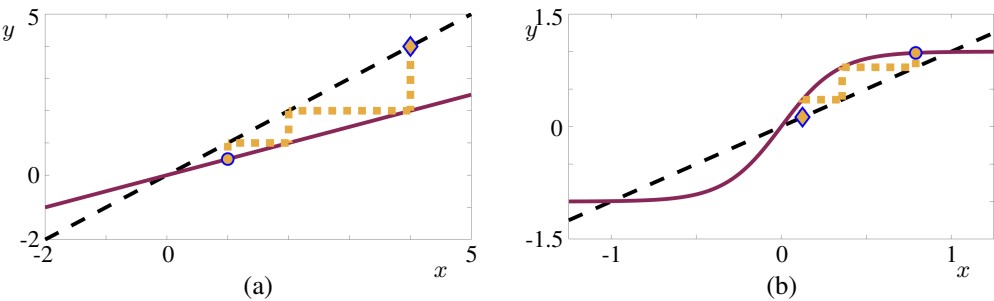

Figure 1: Visualisations of difference equations variable propagation

The dynamic regimes of $s_p$ and $s_q$ behave analogously to the respective difference equations of
**(a)**: $x_t = T(x_{t-1}) = 0.5 x_{t-1}$ and **(b)**: $x_t = T(x_{t-1}) = \tanh(3 x_{t-1})$.
The propagation, in dots, start from rhombuses and end on circles. The dashed lines represent $y = x$ and the solid lines denotes the characteristic outlines of $y = T(x)$. Vertical mappings from $y = x$ to $y = T(x)$ reflect instantaneous variable computations of $T(x_{t-1})$; and horizontal mappings from $y = T(x)$ to $y = x$ represent the recursive nature of difference equations to re-insert the last output as the new input $x_t = T(x_{t-1})$. The origin of **(a)** is a sink which attracts propagation and diminishes variable magnitudes; that of **(b)** is a source which repels propagations and increases variable magnitudes.

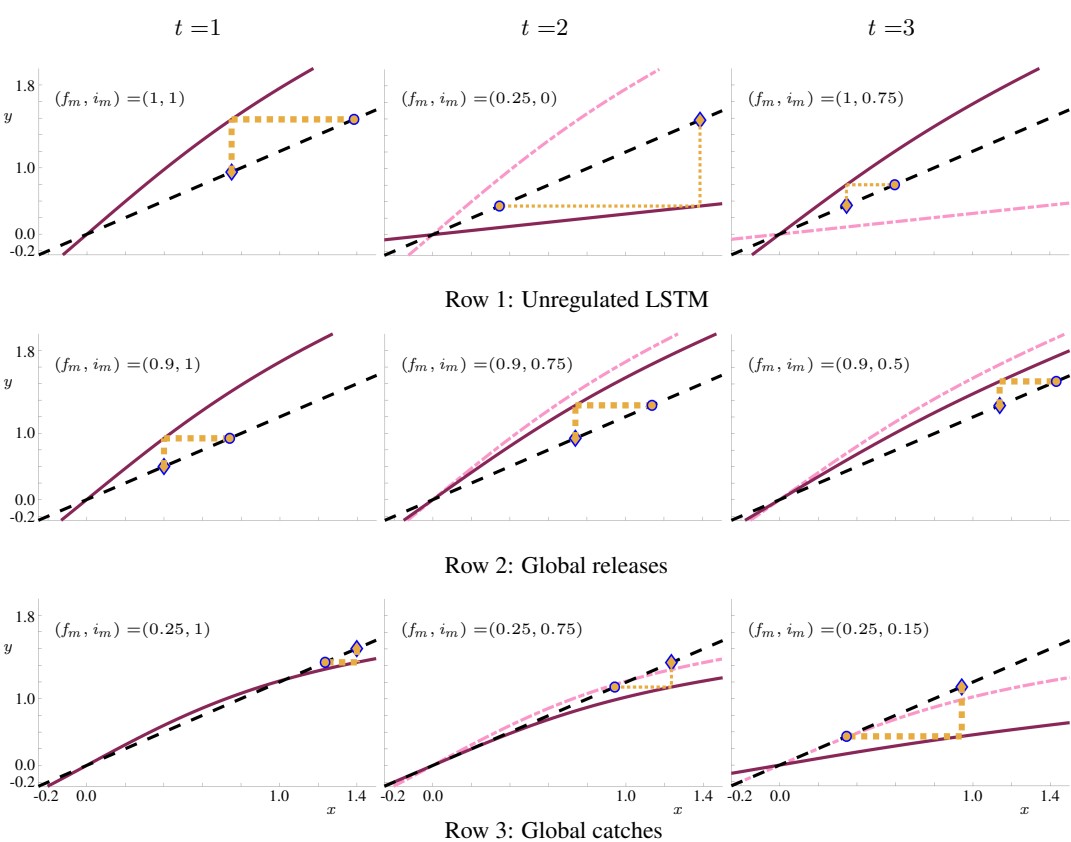

Figure 2: Cell state propagation of controlled and uncontrolled forget gates

Three qualitatively different scenarios for cell state propagation are investigated.
The subfigures follow the presentation format of **Figure 1**; additional dot-dash lines are appended as the characteristic shape of the $t-1$th instance. The characteristic shape of interest is the update of the cell state $y = T(x) = f_{mt} x + i_{mt} \tanh(x)$. From top to bottom, the rows represent for cell state dynamics under an unregulated LSTM, under a large constant forget gate, and under a small constant forget gate, respectively. The latter two scenarios exhibit monotonic propagation because the constant forget gates expose cell states to homogeneous dynamical regimes. The motions are thus more interpretable than those of the unregulated LSTM, where abrupt shrinkage and sudden growth are observed.

## 4 DECAYNETS

This section presents DecayNets[1], and calibrate cell state dynamics according to mathematical insights therein **Subsection 3.3**. From **Figure 2**, it is evident that forget gates deprived of random evolution yield predictable and stable cell state propagation. For this reason, we propose the idea to hard-wire forget gate outputs with a continuous and monotonic decrease.

The DecayNet modifies the forget gate according to

> **System (2):** The Decay mechanism on the LSTM forget gate
>
> the forget-polar input: $\quad \boldsymbol{p}_t = \quad \boldsymbol{p}_{t-1} - \frac{\pi}{2D}\frac{1}{6}(\mathbf{W}_F\boldsymbol{x}_t + \mathbf{W}_{RF}\boldsymbol{q}_{t-1} + 3)$
> purposely presented this way
>
> with $\boldsymbol{p}_0 = \frac{\pi}{2}\cdot\tilde{\mathbf{1}}$,
>
> forget gate elements: $\quad \boldsymbol{f}_{t_i} = \quad \Phi(\boldsymbol{p}_{t_i}) = \begin{cases} 1 & \text{for } \boldsymbol{p}_{t_i} > \frac{\pi}{2} \\ 0 & \text{for } \boldsymbol{p}_{t_i} < 0 \\ \sin(\boldsymbol{p}_{t_i}) & \text{otherwise} \end{cases}$
>
> for $i$ in $1, 2, \ldots N$,

and inherits all remaining variables and their corresponding dimensionalities from **System (1)**. Note, no new learnable parameters are introduced to this system.

The forget gate is initialised as a vector of ones, with equivalent portions of the said initial ones set to be lost over the iterative instances of the LSTM. That is, we set

$$\boldsymbol{f}_{0_i} = \sin(\boldsymbol{p}_{0_i}) = \sin(\tfrac{\pi}{2}) = 1,$$

and that, over time, we remove a maximised cumulative amount of 1, and a minimised cumulative amount of 0, to constrain the final value $\boldsymbol{f}_{t_D} \in [0, 1]$.

We implement the sinusoidal function as the "activation function" of the forget gate. In addition, we engineer "pre-activated neurons" with values within the bounds of $[0, \frac{\pi}{2}]$. Let us elaborate.

We replace $\sigma$ with sin to decay equivalent portions over time. The available amount of instantaneous decay needs to be uniform throughout time to ensure that all instantaneous lost in values are equally important. The sigmoidal outline of $\sigma$ yields less changes when magnitudes of pre-activated values are extreme. We remove this undesirable property with the substitution of sin with radian-like inputs.

Elements of the pre-activated neurons are in the bound of $[0, \frac{\pi}{2}]$. Empirically, we observed that distributions of $\mathbf{W}_F\boldsymbol{x}_t + \mathbf{W}_{RF}\boldsymbol{q}_{t-1}$, with entries of $\mathbf{W}_F$ and $\mathbf{W}_{RF}$ initialised with uniform sampling within the interval of $\pm 1$, are bounded within the limits of $[-3, 3]$.[2] That is, we have

$$\mathbf{W}_F\boldsymbol{x}_t + \mathbf{W}_{RF}\boldsymbol{q}_{t-1} \qquad \in [-3, 3] \text{ and}$$
$$\frac{\pi}{2}\frac{1}{6}(\mathbf{W}_F\boldsymbol{x}_t + \mathbf{W}_{RF}\boldsymbol{q}_{t-1} + 3) \qquad \in [0, \tfrac{\pi}{2}].$$

The design in mind is thus, for each instance, a maximum amount of $\frac{\pi}{2D}$ is lost from $\boldsymbol{p}_{t_i}$'s initialised $\frac{\pi}{2}$. In other words, over an input-length of $D$ units of time, an accumulation, minimised at 0 and maximum at $\frac{\pi}{2}$, is lost in the forget-polar input. The former induces the final value of $\boldsymbol{p}_{D_i} = \frac{\pi}{2}$ (equivalently $\boldsymbol{f}_{D_i} = 1$), and the latter gives $\boldsymbol{p}_{D_i} = 0$ (equivalently $\boldsymbol{f}_{D_i} = 0$).

DecayNets learn to dichotomise cell state values conditioned on the input shown. One subset of the cell state maintains large magnitudes, whereas the other decreases to small magnitudes. The former subset corresponds to dimensions of the forget gate that maintain their large magnitude over time; whereas the latter subset corresponds to those that diminish.

More importantly, the monotonic decrease in the forget gate serves as an interpretation to LSTM's inferential decision mechanism. This is because the update of the cell state, $\boldsymbol{s}_t = \boldsymbol{s}_p + \boldsymbol{s}_q = \boldsymbol{f}_t \odot \boldsymbol{s}_{t-1} + \boldsymbol{i}_t \odot \boldsymbol{a}_t$, is composed of recurrent memory $\boldsymbol{s}_p$ and feedforward memory $\boldsymbol{s}_q$. The irreversible decay indicates that, dimensions of the cell state, that correspond to forget gates that diminish, gradually lose their recurrent nature over time, and effectively function as feedforward neurons. Large forget gate values thus serve as clear indicators for, those neural dimensions, that the DecayNet prefer to reiterate, for the input-driven optimisation.

---

[1] Implementation of our algorithm will be available at https://github.com/anonymous_author/decaynet/
[2] New/revised text are in brown to assist the reviewers, and will be resumed to black after the revision period.

## 5 RELATED STUDIES

The forward pass of LSTMs (and RNNs) consists high dimensional variables with coupled relationships; and is mostly understood with observational studies and search studies. The former explores practical network usages, and the latter finds optimal combinations of network variables.

Observational studies suggest that cell states develop specific functionality. In text (Karpathy et al., 2015), they are observed to act as line length counters and quotation machines; whereas in speech (Wu & King, 2016), they are highly correlated to the Mel-Cepsrtal coefficient. Search studies (Greff et al., 2017; Chung et al., 2014; Jozefowicz et al., 2015) found that, no particular combinatory sets of variables yield an architecture, that consistently outperforms LSTM in all experimental conditions.

Though rigorous and informative, these studies are based on *exhibited qualities of optimised LSTMs*. Karpathy et al. (2015) conjectured the functionality of trained LSTM cell states through examining post-activated cell state values $\tanh(\boldsymbol{s}_t)$. Greff et al. (2017) observed that gated recurrent units (GRUs) (Cho et al., 2014) are capable of performing well on tasks unoptimised by those LSTMs deprived of output gates. Hence, they conjectured that GRUs' coupling of the input gate and the forget gate, and the output gate of LSTMs prevent unbounded cell states. In contrast, our theoretical mathematical approach therein **Sections 3** and **4** are conducted prior to the training of LSTMs.

Mathematical properties of neural networks are mostly studied through the lens of statistical learning theory. This paper took an alternative approach. We treated neural networks as dynamical systems, and offered the perspective of understanding LSTMs as systems of difference (discrete) equations.

Statistical studies and dynamical studies are complementary. Burger & Neubauer (2003) took an inverse problem approach and integrated Tikhonov regularisations to enhance neural network stability. Tallec & Ollivier (2018) offers another theoretical study on the forward pass. Incremental updates of LSTMs are treated as differential equations, and the effects of multiple scales are analysed.

The mathematical analyses therein this paper differs to existing literature because, we propose modifications on *a specific* variable, to yield an *affirmative / deterministic* outcome – DecayNets have *monotonically decreasing forget gates*. In addition, DecayNets initialise their forget gate values as 1s, which is similar to the effect of the common practice of setting LSTM forget gate biases to 1s (Gers et al., 1999), for yielding large initial forget values and for "remembering more by default".

A *hard* regularisation emerges from the reformulation. As mentioned in **Section 4**, the network learns to *irreversibly* suppress the recurrent nature of a selection of cell state values conditioned on the input shown. This differs to soft regularisations such as Dropout (Srivastava et al., 2014) and Zoneout (Krueger et al., 2016), where the respective nullified and frozen activation are recoverable.

## 6 EXPERIMENTS AND RESULTS

### 6.1 PERMUTED SEQUENTIAL MNIST IMAGE CLASSIFICATION

We apply DecayNets and LSTMs on image classification with sequential inputs. The data of choice is MNIST (LeCun et al., 1998), a database of handwritten digits of numbers [0-9], with 60K and 10K images for training and testing respectively. MNIST images are in $28 \times 28$-pixel formats, we present each pixel, one at a time, as a $784(= 28 \times 28)$-unit pixel-by-pixel sequence to the RNNs therein this subsection. Following Le et al. (2015), we apply a fixed random permutation over the pixels for setting up the task commonly known as permuted sequential MNIST (Perm-SeqMNIST).

RNNs of this subsection have a single layer of 100 hidden dimensions and are trained on the RM-SProp optimiser (Tieleman & Hinton, 2012) over 150 epoches. The learning rate is 0.001, with 0.9 momentum and no weight decay. Gradient clipping is applied at 1 to avoid exploding gradients (Pascanu et al., 2012). All weights are initialised with uniform sampling within the interval of $\pm 1$, and all biases are initialised as zeros. No normalisation is performed on the data prior to training, and a softmax classifier is attached to produce prediction from the final hidden state $\boldsymbol{q}_D$.

The best prediction accuracy for unregulated RNNs is 92.1% for DecayNet and 89.5% for LSTM (which roughly matches the baseline model of 89.8% in Krueger et al. (2016)). We then integrated recurrent batch normalisation (RBN) (Cooijmans et al., 2016) to DecayNet and found that both the best performing RBN DecayNet, 97.8%, and the average simulatory performance, 97.4%, yield

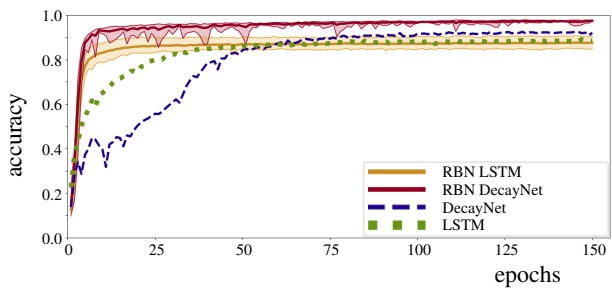

Figure 3: Testing accuracy of the RNNs on Perm-SeqMNIST

Purple and blue denotes DecayNet, while green and yellow is used for LSTM. Performances of RBN RNNs are thick solid lines over shades. Bottoms and tops of the shades are the worst and the best performance of all simulations; and the thick central line is the average performance.

Table 1: Existing competitive results on Perm-SeqMNIST

| Model | Accuracy (%) | Source |
|---|---|---|
| RBN DecayNet-LSTM | Best: **97.8** ; Mean: **97**.4 | This paper |
| Temporal Convolutional Network | 97.2 | Bai et al. (2018) |
| RBN Zoneout LSTM | 95.9 | Krueger et al. (2016) |
| 3D tLSTM+CN | 95.7 | He et al. (2017) |
| Dilated GRU and Dilated CNN | 94.6 and 96.7 | Chang et al. (2017) |
| iRNN | 82.0 | Le et al. (2015) |

better results than the state of the art performance reported in Bai et al. (2018) at 97.2%. **Table 1** presents a list of competitive results in existing literature, and **Figure 3** shows the test set accuracy. Purple and blue denote DecayNets, while yellow and green are used for LSTMs. The best unregulated networks are dashed and dotted; and 10 simulations of RBN RNNs are presented as thick solid lines on shades. Bottoms and tops of the shades are the worst and best performances of all simulations; the central lines are the mean performances. The RBN LSTM results differ to those reported in Cooijmans et al. (2016) and Krueger et al. (2016), we speculate this as a result of differences in the initialisations of weights and biases.

A similar task, of image classification with row-by-row inputs, is presented in **Appendix C**. There, DecayNets outperform LSTMs with less significant differences, but yield more consistent results.

## 6.2 LSTMs as optimisers

The second task follows Andrychowicz et al. (2016) and update weights of multi-layer perceptrons (MLPs) with LSTMs. The paper showed that MLP-optimisees trained with LSTM-optimisers yield marginally lower losses than those trained with the hand-crafted optimiser of ADAM (Kingma & Ba, 2014). Here, we show that the Decay mechanism helps the already successful LSTM-optimiser decrease MLP losses more rapidly, and converge to lower and less varied MLP losses.

We train two-layer MLPs for MNIST classification and update their weights $\boldsymbol{\theta}$ with LSTMs. The MNIST images are normalised with mean and standard deviation (MSD) = $(0.1037, 0.3081)$, and are organise as vector inputs of dimensionality 784. The first MLP layer has weights of dimensionality $\mathbb{R}^{784 \times 20}$; the second layer, a softmax layer, has weights of dimensionality $\mathbb{R}^{20 \times 10}$. The update

$$\boldsymbol{\theta}_t = \boldsymbol{\theta}_{t-1} - \boldsymbol{\alpha}_t \nabla_{\boldsymbol{\theta}_{t-1}} L_t$$

uses learning rate $\boldsymbol{\alpha}_t$ and $\nabla_{\boldsymbol{\theta}_{t-1}} L_t$, the gradient of the optimisation loss with respect to parameter $\boldsymbol{\theta}_{t-1}$. We use LSTM-optimisers to infer $\boldsymbol{\alpha}_t$ given the element-wise product of $\boldsymbol{\theta}_t$ and their corresponding gradients $G_{\boldsymbol{\theta}_{t-1}}$. That is, for each instance, in brevity,

$$\boldsymbol{\alpha}_t, \boldsymbol{q}_t, \boldsymbol{s}_t = \text{LSTM}(\boldsymbol{\theta}_{t-1} \odot G_{\boldsymbol{\theta}_{t-1}}, \boldsymbol{q}_{t-1}, \boldsymbol{s}_{t-1}).$$

The optimisers are trained according to Andrychowicz et al. (2016), which we reiterate in **Appendix D**. For justifications and details on the choices of the input and the output, see **Appendix E**.

The Decay mechanism allows LSTM-optimisers to update their hidden variables for multiple steps before they create learning rates. At the beginning of each instance, the forget polar input is reset as

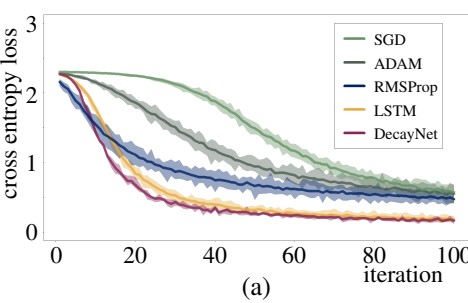 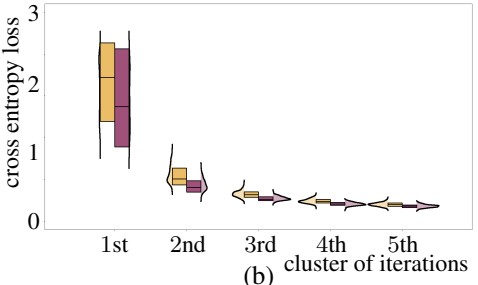

Figure 4: Performances of optimisers

We compare various optimisers to update the weights of MLP-optimisees for learning MNIST classification. Subplot **(a)** shows that, both LSTM-based optimisers, yellow for LSTMs and purple for DecayNets, yield MLPs with marginally lower losses than their rival hand-crafted optimisational schemes. Subplot **(b)** shows that the appended Decay mechanism allows LSTMs to decrease MLP losses more rapidly.

the maximal values of $\frac{\pi}{2} \cdot \tilde{\mathbf{1}}$ (effectively $\boldsymbol{f}_t = \tilde{\mathbf{1}}$). For each instance, the input of $\boldsymbol{\theta}_{t-1} \odot G_{\boldsymbol{\theta}_{t-1}}$ monotonically decrease the forget gate and update the hidden variables as according to

$$\boldsymbol{q}_{t-\frac{1}{2}}, \boldsymbol{s}_{t-\frac{1}{2}} = \text{DecayNet}(\boldsymbol{\theta}_{t-1} \odot G_{\boldsymbol{\theta}_{t-1}}, \boldsymbol{q}_{t-1}, \boldsymbol{s}_{t-1}) \text{ and}$$
$$\boldsymbol{\alpha}_t, \boldsymbol{q}_t, \boldsymbol{s}_t = \text{DecayNet}(\boldsymbol{\theta}_{t-1} \odot G_{\boldsymbol{\theta}_{t-1}}, \boldsymbol{q}_{t-\frac{1}{2}}, \boldsymbol{s}_{t-\frac{1}{2}}).$$

That is, the learning rate is created only after the gated units and hidden variables propagated for two times. Such an internal propagation can be pro-longed, for example, to twenty steps.

A comparison of optimiser performances is presented in **Figure 4(a)**. The figure shows MLP losses after every update of a size 32 mini-batch. The colours of green, grey, blue, yellow, and purple are used to denote stochastic gradient descent with learning rate 0.05, ADAM with learning rate 0.002, RMSProp with learning rate 0.001, a trained LSTM-optimiser, and a trained DecayNet-optimiser, respectively. LSTM-based optimisers outperform hand-crafted rivals by a significant margin, and their MLP losses converge after the 60th update. Hence, we treat losses from the 80th to the 100th update as losses of optimised MLPs. Using these data, two-independent sample t-test found statistical significance that DecayNet-optimised MLPs yield **less** losses than LSTM-optimised MLPs, with MSD= $(-0.13, 0.01)$ by p-value $< 0.001$. In addition, we present **Figure 4(b)**, a sequence of violin-plots, to examine the running performance of the LSTM-based optimisers. With colours inherited from **4(a)**, each violin-plot aggregates twenty iterative updates of loss data, *i.e.*, the first is of the 1st to the 20th update, and that the second is of the 21st to the 40th update. The figure indicates that DecayNet-optimisers minimise MLP losses more rapidly, and converge to less varied losses.

## 7    CONCLUSION

This paper proposes DecayNets, LSTMs with monotonically decreasing forget gates. Motivations and modifications are based on theoretical mathematical clarifications of the LSTM cell state mechanics. The update of the cell state is studied as a difference equation, and we have showen that the update alternates between a catch and a release phase to control shrinkage and growth in its values. We have found that the dynamic alteration is predominately controlled by the forget gate and propose to stabilise the cell state through a monotonically decreasing forget gate. The reformulation is easy to implement and introduces no extra learnable parameters. With RBN, DecayNets achieved better results than previously reported state of the art performance on the Perm-SeqMNIST task. Our experimental results have shown that the Decay mechanism also functions as a promising additional feature for those LSTMs casted as optimisers to update optimisee MLP weights.

However, the most important quality of the Decay mechanism is that it is deterministic. It would be interesting to treat the gradually decaying patterns as features to analyse interactions among speech utterances and other natural language processing tasks (Suominen, 2014). Such a similar deterministic-ness may be hard-wired to GRUs. However, the update of the GRU cell state is more complicated than its LSTM counterpart, and separate dynamical analyses must be conducted.

LSTMs are still opaque. The authors of this manuscript believe that more mathematical analyses should be focused on component $s_q$ of the cell state. We believe this less controllable, and more diverse sub-dynamic of the cell state controls bifurcations (Glendinning, 1994) in LSTM dynamics.

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

## A  A BRIEF INTRODUCTION TO COBWEB DIAGRAMS

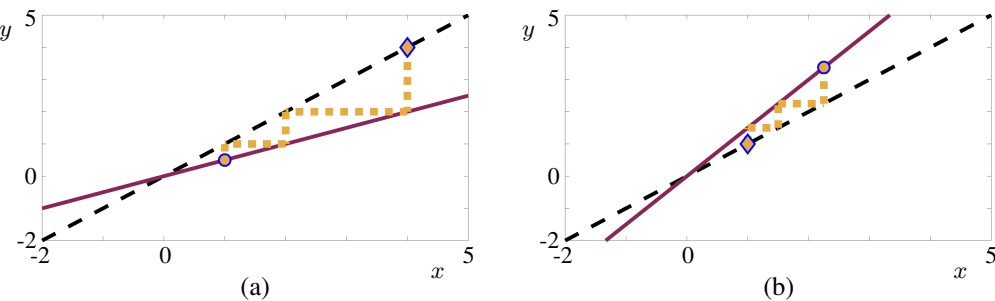

Figure 5: Simple examples of cobwebs

Two simple examples for cobwebs are presented.

Subplot **(a)** is for $y = 0.5x$, while **(b)** is for $y = 1.5x$. Both scenarios share the origin as an equilibrium. Gradients around the vicinity of the equilibria serve as major influences that dictate their stability. Solutions, the neural trajectories, start from the rhombuses and ends at the circles. The gradients around the origin of **(a)** is $< 1$, the equilibrium is hence a sink and attracts neural trajectoreis; its counterpart in **(b)** is $> 1$, hence a source, and repels trajectories. The attractiveness and repulsiveness of the dynamic singularities dictate the growth and shrinkage in variables.

This appendix should only serve as a brief introduction to cobweb diagrams. Interested readers should consult Glendinning (1994) for a broad and complete view on cobweb diagrams, and other dynamical techniques / implications.

Cobweb diagram serves as an intuitive technique to visualise dynamical motions for variables in low dimensional difference equations. Note, not differential equations. Let us first revisit the concept of difference equations, and then introduce cobweb diagrams.

The simplest form of a difference equation is $x_t = T(x_{t-1})$ with $x_0 = \alpha$. Difference equations are simple recursive networks where variable $x_t$ serves as both the input and the output. Subscript $t$ represents computational instances, and $T$ is some arbitrary function.

A cobweb diagram consists 4 fundamental elements. The 45-degree $y = x$ line, the characteristic shape of the operator $y = T(x)$, an initial point $x_0 = \alpha$, and a trail of propagation trajectories. **Figure 5** will be used to support the narrative therein this appendix. The dashed line is $y = x$, the solid line is $y = T(x)$, the initial positions are in rhombuses, the solution trajectories are dotted, and the final positions are in circles.

Cobwebs are useful because they visualise repeated insertions of functional outputs. The repetition is visualised by two qualitatively different types of mappings: a vertical mapping, and a horizontal mapping. Vertical mappings initiate from $y = x$ and terminate on $y = T(x)$, whereas horizontal mappings start from $y = T(x)$ and end on $y = x$. Vertical mappings represent for functional computations, *i.e.*, given the input of $x_{k-1} \to$ compute for the output of $T(x_{k-1})$. Their counterparts of horizontal mappings represent for system updates, *i.e.*, given the post-computational $T(x_{k-1}) \to$ update the network variable of $x_k = T(x_{k-1})$. This is the reason why the variable trajectories shown in **Figure 5** are zig-zag-like. Not all trajectories are zig-zagged, interested readers can find descriptions of tent map propagation in Boeing (2016).

Another important feature to cobwebbing is the labelling of the dynamic singularities known as equilibria. Equilibria are the intersections of $y = x$ and $y = T(x)$. These are the positions where, once a trajectory maps into, or starts on them, the said trajectory will never leave. Trajectories move towards attractive / stable equilibria known as sinks, and moves away from repulsive / unstable equilibria known as sources. For instance, the origin of **Figure 5(a)** is a stable equilibria, and that of **5(b)** is an unstable equilibria. While travelling towards the sink, the variable of **5(a)** decreases in magnitudes, i.e., magnitudes of the $y$-coordinates of the trajectories decrease; and while travelling away from the source, the variable of **5(b)** increases in magnitudes. From these two examples, it is clear that the stability of the equilibria are highly dependent on the gradients around their vicinities. The gradient around the origin of **5(a)** and **5(b)** are $< 1$ and $> 1$ respectively. More interesting dynamics may emerge with more complicated outlines of $y = T(x)$.

## B   A JUSTIFICATION ON THE DEMONSTRATIVE PARAMETERS

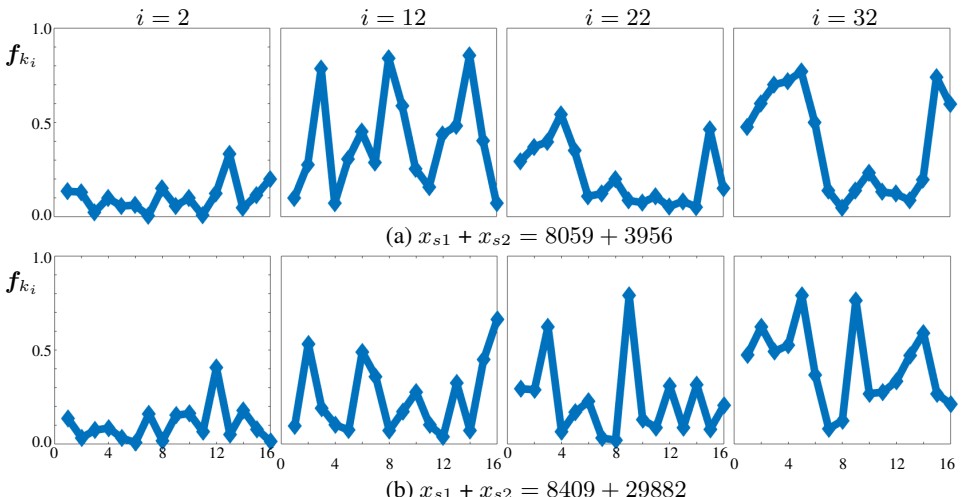

(a) $x_{s1} + x_{s2} = 8059 + 3956$

(b) $x_{s1} + x_{s2} = 8409 + 29882$

Figure 6: Time series of $\boldsymbol{f}_t$ in an optimised LSTM

We insert inputs $x_{s1}$ and $x_{s2}$ as binary strings to a trained LSTM for summation. We present the arbitrary 2nd, 12th, 22nd, and 32nd dimensions of the forget gate during the computation of the LSTM cell. The time series of all of these arbitrary dimensions exhibit near-random dynamics, where extreme dynamic alterations with no clear patterns are observed.

This appendix serves as a justification to the chosen demonstrative parameters in the first row of **Figure 2**. The justification involves 3 items: the vanilla LSTM (see **System (1)**), an additive problem, and records of time series of the forget gate.

*The binary string additive problem defined*:

The vanilla LSTM is provided with two input numbers of $x_{s1}$ and $x_{s2}$ in their 16-digit binary form; the network is trained to generate the correct summation output $y_s$, also in the 16-digit binary form. We apply some simple constraints to ensure that values of both the input and the output do not exceed the maximum capacity of the 16-digit binary form. The maximum value of $x_{s1}$ is limited to the rounded value of $\frac{2^{16}-1}{2}$, and that of $x_{s2}$ is limited to the floored value of $\frac{2^{16}-1}{2}$.

The LSTM in this appendix is a one layer network with 2 input dimensions, 32 hidden dimensions, and 1 output dimension. The network is set to run for 16 units of time, and during each instance, one digit from each of the paired input is forwarded to the LSTM. The binary digits are presented from back to forth. For every instance, the output of the network is computed as follows. Matrix multiplication of a learnable feedforward regressive weight is applied onto the current hidden state $\boldsymbol{q}_t$, the product is then activated via $\tanh$, and finally rounded.

The network is trained in gradient descent with backpropagation through time. The learning rate is set as $0.1$, and the network is not regularised by other means.

*Time series of the forget gate*:

After the network is optimised, we randomly generated two pairs of $x_{s1}$ and $x_{s2}$ and inserted them to the network. Time series of the 2nd, 12th, 22nd, and 32nd dimension of the forget gate are shown in **Figure 6**.

Marginal dimensions of the forget gate oscillate in aperiodic and near-random fashions. There are no clear nor repeating patterns, and it is common to have extreme alterations in oscillatory magnitudes. The demonstrative parameters $f_{m(1...3)} = [1, 0.25, 1]$ mimic these behaviours via an initial decrease followed by a sharp increase; the demonstrative parameters are chosen to provide clear illustrations on the impact to $\boldsymbol{s}_t$ from $\boldsymbol{f}_t$, thus the three values are chosen to induce extreme alterations.

**Figure 2** is based on **Figure 6**. The first two columns of the first row demonstrate an abrupt catch through a large decrease in the forget gate value; whereas the latter two columns of the first row demonstrate a sudden release through a large increase in the forget gate value.

## C  ROW-BY-ROW IMAGE CLASSIFICATION

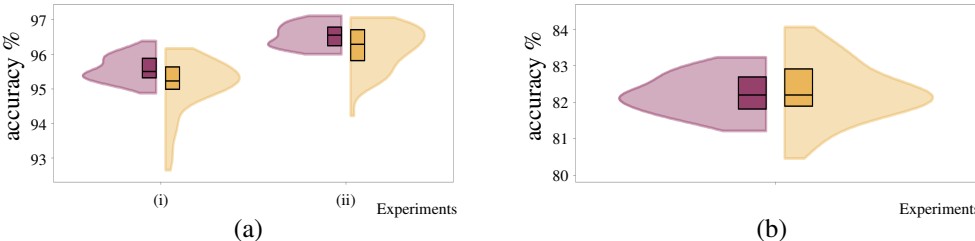

Figure 7: Back-to-back violin plots with side-by-side box plots

Result visualisations, with purple for DecayNet and yellow for LSTM.
Subplot **(a)** is for MNIST; comparisons (i) and (ii) are on 32 and 64 HD networks. The best performing DecayNet outperforms the best performing LSTM; and DecayNets have more consistent results than LSTMs. Subplot **(b)** is for 48 HD DecayNets vs 64 HD LSTMs on Fashion-MNIST. The best performing LSTM outperforms the best performing DecayNet; but arbitrary performances of DecayNets are comparable with LSTMs, and are more consistent.

Table 2: Results

| Data | Model | | Accuracy (%) | t-test ($p$) | Surpass | Best result (%) |
|---|---|---|---|---|---|---|
| MNIST | LSTM | (32 HD) | $95.1 \pm 0.2$ | - - | - - | 96.2 |
| | DecayNet (32 HD) | | $\mathbf{95.6 \pm 0.1}$ | $< 10^{-4}$ | $\checkmark$ | 96.4 |
| | LSTM | (64 HD) | $96.2 \pm 0.2$ | - - | - - | 97.0 |
| | DecayNet (64 HD) | | $\mathbf{96.5 \pm 0.1}$ | $< 10^{-4}$ | $\checkmark$ | 97.1 |
| Fashion-MNIST | LSTM | (64 HD) | $82.3 \pm 0.3$ | - - | - - | 84.1 |
| | DecayNet (48 HD) | | $82.2 \pm \mathbf{0.2}$ | 0.5197 | - - | 83.2 |

We apply DecayNets and LSTMs for image classification with sequential inputs. The data of choice is, again, MNIST, and Fashion-MNIST (Xiao et al., 2017). Fashion-MNIST contains images of fashion products from 10 categories; it also has 60K and 10K images for training and testing, and has images in $28 \times 28$-pixel formats. Comparative results are listed in **Table 2**.

This appendix considers the task of processing images row-by-row, sequentially, from top to bottom. There are thus 28 instances and the inputs are vectors of 28 dimensions. Prior to training, no normalisation is applied to MNIST, and a normalisation of mean and standard deviation
$(\text{MSD}) = (0.5, 0.5)$ is applied to Fashion-MNIST. No other means of regularisation is applied. RNNs of this subsection have one hidden dimension, which we will elaborate later. A softmax classifier is attached to produce prediction from the final hidden state $\boldsymbol{q}_D$; and all RNNs are trained on the ADAM with learning rate 0.01. Note, the term MSD will reappear later.

We conduct two comparisons on DecayNets and LSTMs. The first under the setting of 32 hidden dimensions (HD) and the second with 64 HD. All networks are trained on MNIST for 1 epoch.

Within each HD-group, the best performing DecayNets outperform LSTMs. The results, in the prediction accuracy, are, 96.4% vs 96.2% for 32 HD, and 97.1% vs 97.0% for 64 HD. Two independent sample t-tests show that DecayNets yield **statistically significantly greater** accuracy than LSTMs. For 32 HD models, DecayNets give MSD = (95.6%,0.0037) against LSTMs' MSD = (95.1%,0.0078), with $p$-values $< 10^{-4}$. For 64 HD models, DecayNets give MSD = (96.5,0.0033) against LSTMs' MSD = (96.2%,0.0064), with $p$-values $< 10^{-4}$. We present back-to-back violin plots with side-by-side box plots in **Figure 7(a)**. Purple is used to denote DecayNets, and yellow is for LSTMs. The box plots show that the inter-quatile ranges of DecayNets are smaller than those of LSTMs, implying that the DecayNets yield more consistent and more reliable results.

For Fashion-MNIST, we compare $48$ HD DecayNets against $64$ HD LSTMs over $1$ epoch of training. The DecayNets of this task have $37.44\%$ less parameters than their rivaling LSTMs.

The best performing LSTMs outperform DecayNets, in the prediction accuracy, by $84.1\%$ vs $83.2\%$. However, two independent sample t-test show that there are **no statistical differences** in the mean accuracy for DecayNets and LSTMs. DecayNets have MSD = $(82.2\%, 0.0054)$ against LSTMs' MSD = $(82.3\%, 0.0084)$, with $p$-values $> 0.5$. The result visualisations in **Figure 7(b)**, again, shows that DecayNets yield more consistent results than LSTMs.

## D    DESCRIPTIONS OF THE LEARNING TO LEARN TASK

The learning to learn task follows the paper of Andrychowicz et al. (2016). This section provides some of our own training details, which reiterates some training details provided in the mentioned paper.

Weights of the MLP-optimisee are updated via
$$\boldsymbol{\theta}_t = \boldsymbol{\theta}_{t-1} - \boldsymbol{\alpha}_t \nabla_{\boldsymbol{\theta}_{t-1}} L_t$$
with LSTM-optimisers used for inferring learning rates $\boldsymbol{\alpha}_t$. We considered two-layer MLPs, with dimensionalities $784 \times 20 \times 10$; which we denote the overall learnable parametric count as $r_H$. All weights are initialised randomly from a uniform distribution with the bounds of $[-0.1, 0.1]$, and the biases are initialised as zero vectors. This controlled initialisation is to ensure all MLP-optimisees start as if they were empty canvases, and for maximising the utility of the LSTM-optimisers.

The input of the LSTM, $\boldsymbol{\theta}_{t-1} \odot G_{\boldsymbol{\theta}_{t-1}}$, is the element-wise product of the last instance of MLP parameters and their corresponding gradients. That is,
$$\boldsymbol{\alpha}_t, \boldsymbol{q}_t, \boldsymbol{s}_t = \text{LSTM}(\boldsymbol{\theta}_{t-1} \odot G_{\boldsymbol{\theta}_{t-1}}, \boldsymbol{q}_{t-1}, \boldsymbol{s}_{t-1}).$$

The inputs and the outputs for LSTM-based optimisers differ to those given in Andrychowicz et al. (2016). Justifications are provided in **Appendix E**.

The dimensionality of the vector input $\boldsymbol{\theta}_{t-1} \odot G_{\boldsymbol{\theta}_{t-1}}$ is the total count of MLP learnable parameters, $r_H$, which is a large constant. In order to simplify training for LSTM-optimisers, the LSTMs have a small feedforward layer, located prior to the computation of the gated units, with a learnable matrix of dimensionlity $\mathbb{R}^{r_H \times 28}$, for the purpose of down-sizing the input. Thus, the LSTM-based optimisers have a 'formal' input dimensions of $28$; in addition, the hidden dimensions are chosen to be $64$.

Two separate layer normalisations are applied in the LSTM-optimisers. One to post-process the input-to-hidden connections of $\mathbf{W}\boldsymbol{x}_t$, and the other to post-process the hidden-to-hidden connections of $\mathbf{W}_R \boldsymbol{q}_t$. Trained LSTM-optimisers unfold every instance to create learning rates for the MLP-optimisees. The learning rates $\boldsymbol{\alpha}_t$ are of dimensionality $\mathbb{R}^{r_H}$, and are created from the matrix multiplication of the LSTM hidden state and a learnable up-size matrix of dimensionality $\mathbb{R}^{64 \times r_H}$.

The LSTM-based optimisers are trained via ADAM with learning rate $0.001$. Each newly initialised LSTM-based optimiser learns to update $5$ controlled initialised MLP-optimisees. Weights of the MLP-optimisees are updated for $100$ consecutive instances. For each update, a size $32$ mini-batch is provided to the MLP-optimisees, with records of the differences, between the cross entropy losses, prior and after, the update of MLP-optimisee weights, recorded. For every $20$ update instances, of the $100$ consecutive instances, from the optimisers to the optimisees, an accumulation of the differences in the recorded losses are used to update weights therein the LSTM-based optimisers.

## E    JUSTIFICATIONS ON OUR CHOICES OF INPUTS AND OUTPUTS FOR THE LEARNING TO LEARN TASK

*Justification on inputs*:

Gradients $G_{\boldsymbol{\theta}_{t-1}}$ of the learning to learn task are the products of the pre-processed gradient provided in Andrychowicz et al. (2016). In that paper, the original gradient is given as $\nabla$, and the transformations are given as
$$\nabla^t \rightarrow \begin{cases} \left( \ \frac{\log(|\nabla|)}{10}, \ \text{sgn}(\nabla) \ \right) & \text{if } |\nabla| \geq e^{-10}, \\ \left( \ -1, \quad e^{10}\nabla \ \right) & \text{otherwise.} \end{cases}$$

The authors of this manuscript understood the first pair of gradients $(\frac{\log(|\nabla|)}{10}, -1)$ as an emphasis on large magnitude gradients. Its counterpart pair of $(\mathrm{sgn}(\nabla), e^{10}\nabla)$ provides an emphasis on small magnitude gradients. The first pair consistently produces negative values, whereas the second pair has values of either signs, which mostly dependent on their larger elements. That is, the sign of the second pair could serve as cues, for the directions, to travel sensibly, in search for the solution space for MLPs. In addition, the magnitude of the first pair could serve as cues for the magnitudes of the strides in those directions. Thus, it is not necessary to feed the pre-processed gradients as separate inputs to the LSTM-optimisers. This is because the product of the two retains property of both transformations.

As an additional note, the MLP parameters are element-wise-producted with $G_{\boldsymbol{\theta}_{t-1}}$, the element-wise product of the pre-processed gradients, to prevent the learning of redundant dependencies between MLP parameters and their gradients during the training of the LSTM-optimisers.

*Justification on outputs*:

Therein this paper, the outputs of the LSTM-based optimisers are chosen as the learning rates of the MLP weights. This decision is based on two paper – on the original learning to learn paper of Andrychowicz et al. (2016), and on the learning to learn for few-shot learning paper of Ravi & Larochelle (2016).

In Andrychowicz et al. (2016), the authors used LSTM-optimisers to prepare the whole updating content for the weights of the optimisees. However, the output is not simply added towards the existing weights of the optimisees – the LSTM outputs are rescaled by a factor of $0.1$. The authors of this manuscript speculate this as a consequence of activation function saturation. That is, the output values are close to extrema of the activation functions. Furthermore, we speculate this as a consequnece of the high dimensionality of $r_H$. This serves as another reason for our producted-input, for avoiding excessive optimiser input contributions that stem from the redundant synaptically connected dimensionality.

In Ravi & Larochelle (2016), the authors used LSTM-optimisers to prepare for a set of learning rates and a set of "anti-learning rates" for the weights of the optimisees. The anti-learning rates first relinquishes fractions of the existing MLP weights prior to updating the remaining said weights.

We have concerns for both implementations. We want to avoid a potentially saturated output, and we want an updating rule for MLPs that are mathematically sound. For this reason, we choose to only produce the learning rates via the LSTM-based optimisers. Our experimental results, shown in **Figure 4**, closely resemble those in Andrychowicz et al. (2016).

