# OpenReview forum: "DecayNet: A Study on the Cell States of Long Short Term Memories"
_ICLR.cc/2019/Conference_

### Official Review · AnonReviewer1 · 2018-10-24
**motivation and experiment are not convincing enough**

**Rating:** 4
**Confidence:** 4

**Review:**

This paper provide a modification on the classical LSTM structure. Specifically, it reformulate the forget gate with a monotonically decreasing manner, using sinusoidal function as the activation function.

However, both the motivation and experimental results on such modification are not convincing enough.

1. While there are many heuristic guesses in sec3, important supports of these guesses are missed. For example, Figure 2 is designed to provide supports for the claim that we need controlled forget gates.  However, all the values of forget gates and input gates in Figure 2 are manually set as *conceptual observations*, which provides limited insight on what will happen in the real cases. While the reformulation in sec4 is based on the observations in Figure 2, it is important to plot the real cell propagation after the reformulation, and see whether the real observation meets the conceptual observations in Figure 2.
BTW, Plots in Figure 2 only account for LSTMs' propagation within 3 steps, but in real cases there are way more steps.

2. The authors claim monotonic propagation in the constant forget gates is more interpretable than those of the vanilla-LSTM, as no abrupt shrinkage and sudden growth are observed. But it isn't straightforward to get the relations between abrupt shrinkage and sudden growth on forget gates and the expressive power of the vanilla-LSTM. Also, it's hard to say the monotonic propagation is more interpretable because we don't know what's the meaning of such propagation on the behaviors of LSTMs in applications.

3. The reformulation in sec 4, especially the formula for the forget-polar input p_k, looks heavily hand-crafted, without experimental supports but statements such as "we ran numerous simulations", which is not convincing enough.

4. Experiments are applied on MNIST and Fashion-MNIST. While both datasets are not designed in nature for sequential models like LSTMs. There are better datasets and tasks for testing the proposed reformulation.   e.g. sentence classification, text generation, etc.  No explanation on the choice of datasets.  In addition, the difference between vanilla-LSTM and DecayNet-LSTM is small and it's hard to say it isn't marginal. Maybe larger-scale datasets are needed.

5. Lacking of explanation on specific experimental settings. E.g. training all methods for *only one epoch*, which is very different from the standard practice.

6. More qualitative interpretations for real cell states in both vanilla LSTM  and DecayNet-LSTM are needed. Only conceptual demonstration is included in Figure 2.

---

> ### Author Response · Authors · 2018-11-25
> **To AnonReviewer1 (Part 1)**
>
> R1C01:
> " many heuristic guesses in sec3"
>
> Please note that the core idea is to construct a function that decays an equivalent portion of forget gate values for each iterative instance of the LSTM. This specific design is chosen to ensure that all instantaneous decays are of equivalent importance. The resulting form (which may be perceived too heuristic) is merely a bounding mechanism for the sine function. Regarding the sine function, Subsection 3.2.2 provides motivations by analyzing the dynamics of the forget gate.
>
> The novelty of our approach is to address dynamics of LSTMs. The other reviewers also showed support for the rationales behind the study. For example, as pointed out by AnonReviewer3, the particular set-up does not "just randomly 'reset' the gates" and is based on "an analysis of the cell-state updating scheme of LSTM and realizes that it is mainly controlled by the forget gate". AnonReviewer2 echoed this by stating that "the forget gate seemingly having a stronger effect than the (input gate * input) component, and the authors propose to hard-wire the forget gate to produce a continuous and monotonic decrease". We are saying all these to express that the whole design should not be seen as a heuristic crafting. While we have put immense effort in preparing our paper, if the reviewer feels clarifications on some parts can help getting our message across, we will gratefully follow.
>
> With that said, we reworded Section 4 with a statement that reads
> 'The forget gate is initialised as a vector of ones, with equivalent portions of the said initial ones set to be lost over the iterative instances of the LSTM.'
> in hope of clarification.

---

> ### Author Response · Authors · 2018-11-25
> **To AnonReviewer1 (Part 2)**
>
> R1C02:
> " Plots in Figure 2 only account for LSTMs' propagation within 3 steps"
>
> R1C02 and R1C03 both concern Figure 2 - a collection of plots that present conceptual dynamical motions of the LSTM cell state. This figure consists nine subplots, in three rows and in three columns. Each row corresponds to a distinct time series of forget gate values, with behaviours invoked from, row one - uncontrolled forget gate values, row two - a large constant forget gate value, and row three - a small constant forget gate value, respectively. We visualise these through the 3 columns, the 3 time steps, from left to right, for the consecutive forward propagation of time t = 1, 2, 3. (Originally presented as k = 1, 2, 3; see R3C02 for the reason of changing k to t for denoting instances.)
>
> Though one can certainly continue the analysis for a longer time, we feel that three instances are sufficient to illustrate all major dynamics. The purpose of the first row is to address abrupt catches and sudden releases in vanilla LSTM cell states. An abrupt catch, shown in the transition from first row column one to first row column two, is resulted from a large decline in the gradient of the characteristic outline; whereas a sudden release, shown in the transition from first row column two to first row column three, is resulted from a large increase in the gradient of the characteristic outline. The purpose of the second and the third rows is to show that the forget gate act as the most influential component in the update of the cell state, and that dynamical alterations that are hard to interpret can be removed via controlled forget gate values. Subplots in row two show that cell states continue to grow with large constant forget gates; whereas those in row three show that cell states diminish to zero under small constant forget gates.

---

> > ### Comment · AnonReviewer1 · 2018-12-15
> > **Figure 2 is good but not enough**
> >
> > As I said, figure 2 is good for providing conceptual observations. But it is also important to see what happens in the real cases. The authors mentioned the figure 6 in the appendix has shown LSTM in real cases have chaotic behaviors in terms of the forget gates, which leads to more motivation of a figure showing how Decay acts in real cases.

---

> ### Author Response · Authors · 2018-11-25
> **To AnonReviewer1 (Part 3)**
>
> R1C03:
> "all the values of forget gates and input gates in Figure 2 are manually set as *conceptual observations*"
>
> All parametric values are justifiable and are selected with specific purposes. In addition, the studied behaviours are not rare / uncommon scenarios, nor are they representatives of an unsuccessful training. They were, however, sadly, not well studied.
>
> In Subsection 3.3.1, we stated that the first row of Figure 2 follows experimental results shown in Appendix B. In this appendix, we present Figure 6, time series of arbitrary forget gate dimensions. The presented forget gates belong to a LSTM trained to perform binary string addition. The purpose is to show that, for the simplest tasks, LSTMs possess forget gates with values that oscillate in a fashion that is hard to interpret.
>
> Figure 6 illustrates 16-instance-long chaotic time series in arbitrary dimensions of the forget gate. Repeatedly, chaos in time series takes the form of sudden shiftings, of either hills or valleys, from low to high values, and vice versa. Hence, we extract the repeating chaotic nature with the settings as follows. The first two frames of the first row of Figure 2 address changes in cell state dynamics where forget gates, with values selected to, increase sharply from small to large (equivalently, a hill in Figure 6). Then, the latter two frames depict the alternative cell state dynamics under forget gates which, with values chosen to, drop from large to small (equivalently, a valley in Figure 6).
>
> The second and the third rows of Figure 2 serve to show the opposite scenarios of the first row - we fix the forget gates, as either extremely large values, or as extremely small values, throughout time. These settings serve to provide a stark contrast, for the fact, which is acknowledged by AnonReviewer3, for emphasising that "When the forget is large, say 0.9, the input gate can be anywhere in the range [0.5, 1.0] and still produce growth in the cell value. For forget gate = 0.25, an even larger range of input gate values all produce a shrinking cell value."
>
> In order to provide further clarification, we reworded Subsection 3.3 to include the statement of
> 'We use Figure 2 to support these inferences. The first row simulates the near-random dynamics of a vanilla LSTM cell state. We present rows two and three to show stark contrasts, and advantages, of generating predictable cell state propagation with controllable forget gate values.'
>
> R1C04:
> " the formula for the forget-polar input p\_k, looks heavily hand-crafted"
>
> As mentioned in R1C01, all modification therein Section 4, including the forget-polar input, are motivated on the study presented in Subsection 3.2.2. The conceptual visualisation, which the forget-polar input is based on, is assisted by Figure 2.
>
> As mentioned before in R1C01, the core idea is to construct a function that decays an equivalent portion in the forget gate for each iterative instance of the LSTM. This is to ensure each iterative instance is of equivalent importance to the network. In addition, the forget-polar input makes LSTM mechanics more interpretable. AnonReviewer3 commented on this with "Each forget neuron will decrease at a different rate through the processing of a sequence, leading to sections of the cell state which will decay slowly and sections which will decay quickly."
>
> R1C05:
> " both datasets are not designed in nature for sequential models like LSTMs"
>
> Based on your comment, our revised paper now includes the meta-learning task of learning-to-learn (Andrychowicz et al.,2016). The new task cast LSTMs as optimisers to infer the updating rules of MLP-optimisees on MNIST classification. This is a naturally sequential task with an arbitrary length. Details are provided therein the General Comment.
>
> [Andrychowicz et al. (2016)]
> Learning to learn by gradient descent by gradient descent.

---

> > ### Comment · AnonReviewer1 · 2018-12-15
> > **still not a common task for LSTM and decayNet**
> >
> > The task of learning to learn is good, but similar to Reviewer3, this does not occur to me as a common task for sequential models. As suggested by Reviewer3, I think tasks like Penn Treebank are important.
> > Also, similar to Reviewer3, I've some doubts that the statement "DecayNet has more modeling power than LSTM" is generally applicable, and my suggestion is doing more experiments on standard tasks, or adding analysis that when will this statement hold and when will not, which could potentially increase the influence of this idea.

---

### Official Review · AnonReviewer3 · 2018-10-31
**Unclear general utility, insufficiently explained experiments.**

**Rating:** 4
**Confidence:** 4

**Review:**

This paper analyses the internal dynamics of an LSTM, focusing on the cell state as being the most important component, and analyses what directly influences the contents of the cell state using difference equations. The authors note that at any timestep, the output cell state is the sum of (previous cell state * forget) and (input gate * input). The former can only shrink or maintain the cell value, which the authors label 'catch' and the latter can increase the magnitude, labelled 'release'.

The authors show that for a single neuron, with chosen values for the forget gate and inputs, consistent growth or consistent shrinking of the cell state can be observed. When the forget is large, say 0.9, the input gate can be anywhere in the range 0.5, 1.0] and still produce growth in the cell value. For forget gate = 0.25, an even larger range of input gate values all produce a shrinking cell value.

Due to the forget gate seemingly having a stronger effect than the (input gate * input) component, the authors propose to hard wire the forget gate to produce a continuous and monotonic decrease, producing the DecayNet. The rate of this decay is controlled by a learned function of the input and previous hidden state, with some shifting in order to maintain a monotonic decrease. Each forget neuron will decrease at a different rate through the processing of a sequence, leading to sections of the cell state which will decay slowly and sections which will decay quickly.

The authors perform two sets of experiments. The second is sequence classification with the standard 'pixel by pixel' permuted sequential MNIST, in which they show a new SOTA with using Recurrent Batch Norm combined with DecayNet. They also demonstrate a DecayNet with fewer parameters producing roughy the same median performance as a normal LSTM but with lower variance.

The first experiment is described as "image classification", with MNIST and Fashion-MNIST. This section is unclear to me, I had initally assumed that the data would be fed in one pixel at a time, but due to the presence of the other experiments I presume this is not the case. It is not clear what the 'time' dimension is in how the RNNs are applied here, if not through some ordering of pixels. If the entire image is presented as a flattened input, and the time dimension is iterating through the dataset, then there is no reason to use an RNN here. More detail must be added here to make it clear exactly how these RNNs are being applied to images - the text says the softmax layer is produced from the final hidden state, but without the information about how the different hidden states are produced for a given training example this not meaningful. I can imagine that both tasks are pixel by pixel, and the only difference is whether to apply the permutation.. but that is my guesswork.

In general I find this paper an interesting idea, reasonably well communicated but some parts are not clear. All the experiments (as far as I can tell) work on fixed length sequences. One advantage of an LSTM is that can run onnline on arbitrary length data, for example when used in a RL Agent. In those circumstances, does learning a fixed monotonic delay on the forget gate make sense? I would guess not, and therefore I think the paper could be more explicit in indicating when a DecayNet is a good idea.

There are definitely tasks in which you want to have the forget gate drop to zero, to reset the state, and then go back up to 1 in subsequent timesteps to remember some new information. Presumably the monotonic delay would perform poorly.

Is DecayNet appropriate only when you have fixed length sequences, where the distribution of 'when does relevant information appear in the input' is fixed? These questions make me doubt the generality of this approach, whereas "this reformulation increases LSTM modelling power ... and also yields more consistent results" from the abstract reads like this is a strictly better LSTM. A much wider variety of experiments would be required to justify this sentence.

It would be interesting to see some diagrams of forget gate / cell state changes throughout a real task, ie a graph with `k` on the x axis. The presentation of the new forget gate in "System 2" is clear in terms of being able to implement it, but it's not intuitive to me what this actually looks like. The graphs I suggest might go a long way to providing intuition for readers.



Overall while I like the spirit of trying to understand and manipulate LSTM learning dynamics I am recommending reject. I do not think the paper sufficiently motivates why a monotonic decay should be good, and while the new SOTA on permuted MNIST is great, I'm concerned that the first experiments are not reproducable, as detailed previously in this review. All hyperparameters appear to be present, so this paper would be reproducable, except for the NIST experiments.


General recommendations for a future resubmission:

* Clarify description of first MNIST experiments, and how they are different from permuted MNIST.
* Experiments on a wider variety of canonical RNN tasks - Penn Treebank is an obvious contender.
* Some mention of in what situations this is obviously not a good model to use (RL?)
* More intuition / visualisations as to what the internal dynamics inside DecayNet look like, vs normal LSTM.
* Devote less space to the initial dynamics analysis, or modify to be representative of a real task. This part was interesting on first read, but the only thing I think it really proves is 'when we artificially choose the input values things get bigger or smaller'. The important thing is, what actually happens when training on a task that we care about - if the same catch and release dynamics are observable, then that makes the idea more compelling.


Notes and suggestions:
I feel the notation would be clearer if instead of k = 1 .. D, this index was t = 1 ... T. This makes it cleare that s_k is not the k'th item in the array, but rather than whole activation array at a specific time. The notation \sigma_t could then be replace with \tanh.

"We replace \sigma_t with sin for an ergodic delay over time": as this is a new gate for the forget gate, should this be \sigma_s?

One DL rule of thumb heard relatively often is to simply initialise LSTM forget gate biases to 1, to "remember more by default". As this is a (much simpler) way of trying to influence the behaviour of the gate, and it anecdotally improves data efficiency, it is worth mentioning in the paper.

---

> ### Comment · AnonReviewer2 · 2018-11-03
> **I agree, the experiments should be described much more clearly**
>
> During my review, I focused more on the theoretical part and less on the experiments. Definitely it is true that the experiments are not described with enough detail. It should be a general manner that paper submissions include all code and experimental parameters in additionally attached files or accessible via a virtual machine. Maybe it is possible for the authors to provide a github link or similar. Could the reproducibility change your decision to the positive side?

---

> > ### Comment · AnonReviewer3 · 2018-11-05
> > **Experiments more important than code release**
> >
> > Clarity on how exactly the non-sequential MNIST experiments work is essential, but to change my opinion to accept I think more experiments are required, at least on something like PTB. I don't think code release / virtual machine should be mandatory, there are often institutional barriers (reliance on in-house libraries etc) which make this extremely costly. More experiments are the key point for me.

---

> ### Author Response · Authors · 2018-11-25
> **To AnonReviewer3**
>
> R3C01:
> "should this be sigma\_s?"
>
> Thank you for your careful eyes with this typo. We have fixed this in our revised paper.
>
> R3C02:
> " The notation sigma\_t could then be replace with tanh."
>
> Based on your suggestion, we have changed the notation to make our work more accessible. We have also changed k, which we initially used to denote for instances, into t .
>
> R3C03:
> " ... to simply initialise LSTM forget gate biases to 1..."
>
> Based on your suggestion, we included this general practice in Section 5 -- Related Studies. The text now reads as
> "DecayNets initialise their forget gate values as 1s, which is similar to the effect of the common practice of setting LSTM forget gate biases to 1s (Gers et al., 2000), for yielding large initial forget values and for 'remembering more by default'."
>
> [Gers et al. (2000)]
> Learning to forget: Continual prediction with LSTM.
>
> R3C04:
> " The first experiment is described as `image classification" and
> " It is not clear what the 'time' dimension is in how the RNNs are applied here ..."
>
> The experiment set-up follows the guidelines of Metz et al. (2016) (see Appendix C). We have now revised the text to make this experiment more accessible. In particular, we have added the explanation of
> "The images are processed row-by-row, sequentially, from top to bottom. Thus, there are 28 instances and the inputs are vectors of 28 dimensions."
>
> As addressed in the General Comment, the first experiment of the original submission is now placed in Appendix C of the revised paper.
>
> [Metz et al. (2016)]
> Unrolled  generative  adversarial networks.
>
> R3C05:
> " A much wider variety of experiments ..." and
> " All the experiments (as far as I can tell) work on fixed length sequences. One advantage of an LSTM is that can run online on arbitrary length data"
>
> We have now added the arbitrary-length task of learning-to-learn (Andrychowicz et al., 2016) in Subsection 6.2 of the revised paper. This challenging task casts LSTMs as optimisers for updating the learning rule for another optimisee neural network. The LSTM-based optimisers are used to aggregate optimisee gradients for updates of arbitrary lengths, until the optimisee losses converge. Please refer to the General Comment for details.
>
> [Andrychowicz et al. (2016)]
> Learning to learn by gradient descent by gradient descent.

---

> > ### Comment · AnonReviewer3 · 2018-11-28
> > **Learning 2 learn experiments not a canonical RNN task**
> >
> > I think the authors for their rebuttal and updated version. However I will be keeping my score the same.
> >
> > The learning to learn task is interesting but I feel it doesn't fall into the category of a canonical RNN task. The authors cite two previous papers who have approached it, but note that the exact problem setup is different to both papers. The authors provide justifications, and this may be reasonable, but it means the task is now not directly comparable to any other papers. Even if the performance of the LSTM baseline is similar to the original L2L paper, this is not in my opinion a canonical RNN task. Language modelling, using exactly the same setup as Melis et al 2017 - https://arxiv.org/abs/1707.05589 (to give one recent example) will make the general utility of this model far easier to assess.
> >
> > Additionally, the L2L performance of LSTM vs DecayNet seems miniscule, especially by the time of convergence.
> >
> > I appreciate the clarification about where the row by row MNIST problem setup comes from. I note the Metz et al citation does not appear in the current draft. Moreover, assuming this is drawing an analogy to section 3.2 https://arxiv.org/abs/1611.02163 - that paper is *generating* mnist column by column, whereas this paper is classifying after reading the data row by row. The column / row difference I'm sure is not important, but the difference between generating and discriminating is significant. Without further justification of why this task setup is interesting in addition to the permuted MNIST results, I'm not sure this section adds to the paper, even as a appendix.
> >
> > Overall I feel the revised paper does not make clear the general utility of this LSTM variant. With good performance on widely used benchmarks this could change, but the changes made in the revision do not satisfy this criteria, in my opinion.

---

### Official Review · AnonReviewer2 · 2018-11-03
**Theoretical Analysis of the forget gate behaviour leading to a nice novel contribution**

**Rating:** 8
**Confidence:** 3

**Review:**

This paper performs an analysis of the cell-state updating scheme of LSTM and realizes that it is mainly controlled by the forget gate. Based on these outcomes they reformulate the functions (interpreted as differential equations) to add a decay-behaviour in the forget gates, finally called DecayNet-LSTM.


The theoretical analysis in this paper is very welcome and goes beyond observations which we made in the past, i.e., we often saw similar behavior in our experiments and as the authors also state in Section 5, there have been previous observations and approaches. In 2016, I have seen an idea called Random Activation Preservation (RAP) (https://ieeexplore.ieee.org/abstract/document/7487778 ) which just randomly "resets" the gates. However, they only show empirical outcomes, not a sophisticated analysis as in this paper.

In the experiments it is shown, that the DecayNet-LSTM performs similarly, or sometimes better than simple LSTM on standard tasks. On more difficult tasks, such as Perm-SeqMNIST, a state-of-the-art performance is achieved.

Minor comments:
Please note, it should be Long Short-Term Memory (with hyphen between short and term)
You call the contribution DecayNet; And in the paper sometimes refer to it as DecayNet-LSTM; Maybe there could also be a DecayNet-GRU, ... If you, instead of "reformulation", would clearly write that DecayNet is an addition to the LSTM architecture, it might be more clear.

---

> ### Author Response · Authors · 2018-11-25
> **To AnonReviewer2**
>
> R2C01:
> "Maybe there could also be a DecayNet-GRU"
>
> The updating scheme in GRUs is different and more complicated, therefore, the DecayNet cannot be applied there verbatim. While modifying GRUs with a deterministic feature similar to the Decay mechanism is a viable research direction, we believe it goes beyond the scope of our current work. Having said this, we have reflected your comment in Section 7 -- Conclusion.
>
> R2C02:
> "instead of 'reformulation', would clearly write that DecayNet is an addition to the LSTM architecture, it might be more clear."
>
> We have rephrased to address your comment in Section 1 - Introduction. We have now noted that
> "We introduce DecayNets as an addition to the LSTM architecture –-- DecayNets have monotonic decays in forget gates and enjoy smooth transitions in their cell state values."

---

### Author Response · Authors · 2018-11-25
**General Comment**

We thank all reviewers, for spending time to review our paper, and for their constructive feedback. We have taken all their comments on board and revised our work accordingly. In order to assist the reviewers with the changes, the new/revised text are in colour brown in the revised-submission; the change of colours will be removed if the paper gets accepted.

We have added a new experiment to our paper by incorporating DecayNets to address a meta-learning problem; see Subsection 6.2 where DecayNets are used for the problem of learning to learn (Andrychowicz et al, 2016). This is to address the comment of AnonReviewer1 and AnonReviewer3 regarding more evidence for naturally sequential tasks where the length of the sequence is not fixed. We have also tightened our language according to the reviewers' comments; we fixed typos and proofread our work. Below, we address the common  concern of AnonReviewer1 and AnonReviewer3 for this additional experiment. Whereas point-by-point responses to reviewers' specific comments can be found as replies to the reviewer's feedback.

AnonReviewer1
"Both datasets are not designed in nature for sequential models like LSTMs."
and
AnonReviewer3
"All the experiments (as far as I can tell) work on fixed length sequences"

Following Andrychowicz et al. (2016),  we cast DecayNets and LSTMs as optimisers to update weights of MLP-optimisees on the task of MNIST classification. This task has an arbitrary length nature  which requires to aggregate the gradient information in a way that the optimizees can be optimized with minimum steps.

The MNIST images, of size 28 x 28, are presented as vector inputs of size 784 (= 28 x 28) to the optimisees. The optimisees are two-layer MLPs with dimensionality 784 x 20 x 10; the second MLP layer is a softmax layer where cross entropy losses need to be minimised.

During the training phase, weights of the optimisees, along their corresponding gradients, are passed as inputs to LSTM-optimisers. For every instance, LSTM-optimisers prepare updating rules for the weights of the optimisees.

The Decay mechanism allows LSTMs to update their hidden variables for multiple times before generating the updating rules. For every instance, DecayNet forget gates are first reset as 1s, then, the inputs monotonically decay the forget gates, and update their hidden variables, for two instances. From our experiments, we found that the Decay mechanism helps the already successful LSTM-optimiser decrease MLP losses more rapidly, and yield optimised MLPs with lower and less varied losses.

In order to make space for the new experiment, we reconstructed Section 6. To be precise, we put
the experiment on image classification with row-by-row inputs in Appendix C of the revised paper.

[Andrychowicz et al. (2016)]
Learning to learn by gradient descent by gradient descent.

---

### Meta-Review · Area_Chair1 · 2018-12-14
**further work needed**

**Confidence:** 3
**Recommendation:** Reject

**Metareview:**

there is a disagreement among the reviewers, and i am siding with the two reviewers (r1 and r3) and agree with r3 that it is rather unconventional to pick learning-to-learn to experiment with modelling variable-length sequences (it's not like there's no other task that has this characteristics, e.g., language modelling, translation, ...)